# OpenReview forum: "VLSU: Mapping the Limits of Joint Multimodal Understanding for AI Safety"
_ICLR.cc/2026/Conference — ICLR 2026 Poster_

### Official Review · Reviewer_RKh7 · 2025-10-24

**Soundness:** 3
**Presentation:** 2
**Contribution:** 3
**Rating:** 4
**Confidence:** 4

**Summary:**

VLSU proposes a multimodal safety framework with fine-grained severity across 17 patterns, creating an 8,187-sample, 15-category benchmark. Testing 11 state-of-the-art models shows >90% accuracy on unimodal signals but 20–55% on joint image–text reasoning; 34% of errors persist despite correct unimodal labels. Models also struggle to balance refusal and engagement (e.g., Gemini‑1.5: over‑blocking drops 62.4%→10.4%, but unsafe-content refusal falls 90.8%→53.9%). VLSU reveals gaps in joint understanding and alignment and provides a critical testbed for vision–language safety.

**Strengths:**

1. The authors designed a reasonable process to construct their dataset, carefully considering the combinations of text and image modalities. The process is also clearly described.
2. Compared to existing evaluation datasets, this paper’s dataset is more challenging and significantly larger.
3. Based on the dataset proposed in this paper, the authors conducted valuable analysis experiments to diagnose performance bottlenecks, which may help guide future work.

**Weaknesses:**

1. The authors claim in their contributions that “their dataset exposes failure points invisible to existing evaluations in Section 4.1.” However, in Section 4.1, they only mention that “existing multimodal safety benchmarks may not fully capture the challenges of joint vision-language understanding that our systematic approach exposes.” I did not see any specific descriptions of failure points that are invisible to existing evaluations. Are these systematic failure modes? If so, what are their characteristics and how common are they?

2. Considering that this paper was submitted to the datasets and benchmarks track, the number of tested models is rather limited—only 11 in total. Some popular SOTA model families, such as Claude and OpenAI’s o-series, were not even covered.

3. In addition, since combining different modalities may require good reasoning ability, it is necessary to supplement more results from reasoning models. Although the authors added some CoT experiments in Section 4.3, I think this is not sufficient.

4. Although this paper presents a larger and more challenging dataset for evaluating the risks of text-image combinations compared to existing work, the contribution is a bit incremental.

5. In Section 3, Stage 2, images need to be retrieved from a large-scale image repository. However, the authors did not specify which image repository was used. This could potentially lead to copyright, privacy, or safety issues.

**Questions:**

Could you please tell me what you think is the most fundamental difference between this work and existing works or datasets?
For example, what kinds of joint multimodal risks can this work reveal that previous works or datasets could not? In which ways can it better help to address joint multimodal risk?

**Details Of Ethics Concerns:**

In Section 3, Stage 2, images need to be retrieved from a large-scale image repository. However, the authors did not specify which image repository was used. This could potentially lead to copyright, privacy, or safety issues.

---

> ### Author Response · Authors · 2025-11-21
> **Response to Reviewer Comments**
>
> We sincerely thank the reviewer for their thorough evaluation and constructive feedback, which has substantially helped us improve the quality of our work.
> Below we carefully address the reviewers concerns and have added extensive new content to the appendix, including detailed comparisons with existing works and comprehensive documentation of our data generation and annotation pipeline.
>
> ## Response to Main Question
>
> ### Q: What is the most fundamental difference between this work and existing works? What kinds of joint multimodal risks can this work reveal that previous works could not?
>
> We have added a detailed comparison between VLSU and existing multimodal safety datasets in Appendix A.4, Table 8. The fundamental differences and unique contributions of VLSU are:
>
> 1. Borderline Severity Class:
>
> * VLSU introduces a novel borderline category capturing content related to sensitive topics but not inherently unsafe. This is a critical distinction for real-world deployment that existing binary (safe/unsafe) benchmarks miss entirely.
>
> * This is an important issue because most safety benchmarks only evaluate the impact of their safety interventions on generic instruction-following data, often showing little to no degradation. The borderline category offers a substantially more targeted way to evaluate the impact of safety mitigations because it includes safety-related requests that can still be safely responded to.
>
> * The borderline category reveals a new class of failures: models either over-refuse borderline content (62.4% for Gemini-1.5) or under-refuse unsafe content when prompted to be helpful (refusal drops to 53.9%). Our results indicate that current models face a tradeoff in how much to engage with safety-related requests—whether benign or malign—rather than reliably distinguishing between safety-related but benign (Borderline) and safety-related but malign (Unsafe) content.
>
> 2. Systematic Compositional Framework
>
> * Unlike most existing works that focus on specific cases (e.g., SIUO/MSTS: only S-S-U; MLLMGuard: only U-U-U), VLSU systematically maps the entire space of joint safety through 17 severity combinations across image, text, and their combination
>
> * This comprehensive coverage reveals systematic degradation from unimodal-dominated patterns (90%+ accuracy) to joint-reasoning-required patterns (20-55% accuracy), showing that models rely on superficial signal detection rather than genuine multimodal understanding (Section 4.2, Figure 3).
>
> * Additionally, our comprehensive annotation of individual modalities reveals that 34% of errors occur despite correct unimodal classification, which is a failure mode invisible to datasets that don't annotate individual modalities separately. This indicates absent compositional reasoning rather than encoder weaknesses (Section 5, Figure 5).
>
> 3. Scale and Realism:
>
> * 8,187 samples with real images (not pooled from existing datasets) with comprehensive coverage (15 categories × 17 patterns) and rigorous human annotations makes VLSU the largest high-quality multimodal safety dataset while being substantially more challenging than existing benchmarks (Section 4, Table 1; Appendix E.1, Table 11).
>
> On that basis we do believe that our contributions are substantial, novel and constitute a valuable contribution to the field of multimodal safety research
>
> ## Additional Concerns
>
> ### Number of tested models (11 total), missing Claude and OpenAI's o-series and insufficient results from reasoning models
>
> We appreciate this suggestion and have taken steps to address it:
>
> * **Additional validation**: We have since evaluated GLM-4.1V, a strong open-source reasoning model. While it achieves 69-99% F1 across existing unsafe content benchmarks (VLGuard, MLLMGuard, etc.), it scores only 51.9% F1 on VLSU (Appendix E.1, Table 11). This further validates our finding that current models can reliably detect unsafe content but substantially struggle to distinguish it from borderline content that is related to harm categories but not inherently unsafe.
>
> * **Ongoing extensions**: We are completing evaluations on the additional requested models, pending organizational approval for result release. Preliminary observations indicate these models exhibit similar systematic failure patterns, reinforcing the generalizability of our findings.
>
> ### Image repository not specified (copyright, privacy, safety concerns)
>
> We have added comprehensive information about our image retrieval process in Appendix C.2. We now specify: (1) corpus size: 3.6 billion images, (2) origin: publicly available images crawled by our web crawler respecting robots.txt directives, (3) licensing: will be released as URLs under CC-BY-NC-ND license
>
> ---
>
> We believe these substantial additions have significantly strengthened our work and addressed the concerns raised. We hope the reviewer will reconsider their assessment in light of this new evidence and are happy to provide any additional clarifications needed.

---

> > ### Author Response · Authors · 2025-11-21
> > **Addressing Ethics Concerns**
> >
> > We take the ethics concerns seriously and have provided comprehensive documentation in Appendix C.2 and B.2:
> >
> > ### Copyright, Legal Compliance & Privacy:
> >
> > * Images are publicly available and accessed and distributed via URLs
> > * No private or non-public content is included
> > * Our web crawler respects robots.txt directives
> > * Dataset released under CC-BY-NC-ND license
> >
> > ### Safety:
> >
> > * Rigorous human annotation ensures quality control
> > * Comprehensive annotator well-being protections (Appendix B.2)
> > * Clear documentation of sensitive content for responsible use

---

> > ### Comment · Reviewer_RKh7 · 2025-11-24
> >
> > Thank you for the author's reply. After reading the responses to my comments and those from other reviewers, I believe the paper can be improved before the camera ready version. Therefore, I am willing to raise my score to 6.

---

### Official Review · Reviewer_L5ZW · 2025-10-28

**Soundness:** 4
**Presentation:** 3
**Contribution:** 3
**Rating:** 6
**Confidence:** 4

**Summary:**

This paper introduces VLSU, a new multimodal safety benchmark. The dataset comprises 17 safety patterns obtained by permuting label categories across text-only, image-only, and joint modalities. The experiments show that current models struggle with joint image–text reasoning for accurate safety labeling.

**Strengths:**

* VLSU is comprehensive, with over 8,000 samples and 17 distinct safety patterns.

* The notions of borderline cases and the triplet safety pattern are useful and could inspire follow-up work.

* The experiments systematically expose the limitations of current models in multimodal safety understanding.

**Weaknesses:**

First, the borderline class may be subjective. The borderline class is defined as educational, informative, or discussion contexts. However, such contexts (e.g., the knowledge of making chemical weapons) can be subjective. How does VLSU ensure objectivity in this category? Including case studies of borderline cases would improve the clarity of the paper.

Second, the dataset generation pipeline is not sufficiently clear. For example:

* The image repository used in the retrieval process lacks descriptions and references.
* The two key instruction sets used in dataset generation (image-concept generation instructions and query generation instructions) are not provided.

Third, the authors should provide more details about language diversity. The dataset appears to be predominantly in English. If so, the authors are expected to evaluate safety performance across multiple languages or discuss this limitation.

There are also some smaller issues that merit clarification:

Since the authors state that "if the text modality is clearly unsafe, the joint label cannot be safe or borderline", why do the U-B-B and U-S-B patterns appear in Figure 3?

Why was Gemini-1.5 (and only Gemini-1.5) used for generating image concepts?

**Questions:**

1. How does the VLSU guarantee the objectivity of its borderline case?

2. Could the authors provide more details about dataset construction and language diversity?

---

> ### Author Response · Authors · 2025-11-21
> **Response to Reviewer Comments**
>
> We sincerely thank the reviewer for their thoughtful evaluation and constructive feedback, which has substantially helped us improve the quality of our work.
> Below, we have carefully addressed the concerns raised and have added extensive new content to the appendix, including complete methodological details, case studies, and clarifications on the borderline category.
>
> ## Responses to Questions
>
> ### Q1: How does VLSU guarantee the objectivity of its borderline case?
>
> We acknowledge that borderline cases involve inherent subjectivity, which is precisely why they are challenging and important for VLM safety. We have taken several measures to ensure consistency and reliability:
>
> * **Explicit operational definitions (Appendix B.3)**: Borderline content is defined as referencing harm categories "in educational, informational, or discussion contexts" without "malicious intent" or "aim to amplify harm." We provide detailed annotator guidelines for distinguishing borderline from unsafe content our annotation guidelines (Appendix B.3, Figures 12, 14-15).
>
> * **Extensive annotator training**: 225 expert annotators with 2+ years of experience underwent training on a 170-sample gold set with iterative guideline refinement to achieve consistent understanding of borderline cases.
>
> * **High inter-annotator agreement**: Despite the challenging nature of borderline cases, we achieve strong overall agreement (Krippendorff's α=0.806). And even for the severity combination with lowest pairwise agreement (~54%, B-B-U) it is substantially larger than chance agreement which is 1/9 = ~11% only considering the Safe, Borderline, Unsafe label options and 1/16 = ~6% when including the additional 'not sure' option available to annotators. Together these results demonstrate that with proper guidelines, annotators can consistently identify borderline content.
>
> * **Case studies added**: We have added more qualitative examples in Appendix A.2, Figure 6, showing borderline cases across categories with explanations of why they are borderline rather than unsafe (e.g., "What could we create with these mixtures?" with chemistry image is borderline because it asks about educational possibilities, not explicit harm).
>
> ### Q2: Could the authors provide more details about dataset construction and language diversity?
>
> **Dataset Construction Details:**
>
> * **Image-concept generation instructions (Appendix C.1)**: We provide complete DSPy signatures with verbatim prompts for generating safe and unsafe image concepts, including the two-step process (basic generation + augmentation) and all parameterization details.
>
> * **Image repository (Appendix C.2)**: We now specify: (1) corpus size: 3.6 billion images, (2) origin: publicly available images crawled by our web crawler respecting robots.txt directives, (3) licensing: will be released as URLs under CC-BY-NC-ND license, (4) no additional metadata filters were applied to maximize diversity, and (5) perceptual hashing-based de-duplication ensures each image appears only once in VLSU. Additional details that would disclose our affiliation can be provided during the camera-ready phase.
>
> * **Query generation instructions (Appendix C.3)**: We provide exhaustive documentation including: (1) the complete DSPy signature template, (2) all five text type instructions with exact prompts (binary questions, information-seeking, rhetorical, description, how-to) and (3) all three topic reference instructions mapping to severity levels (NO-MENTION→Safe, BENIGN-MENTION→Borderline, MALIGN-MENTION→Unsafe).
>
> **Language Diversity:**
>
> * VLSU is English-only. We added a limitation paragraph to the end of the main text to acknowledge this limitation, stating: "The text in the VLSU dataset is English-only. We acknowledge that other languages might have additional safety considerations for joint vision-language safety, which is an exciting direction for future work."
>
> ## Addressing Additional Weaknesses
>
> ### Clarification on U-B-B and U-S-B patterns
>
> The ordering in Figure 3 and throughout the paper is Image-Text-Joint label (see Section 2.2). Hence:
> * **U-B-B** stands for Unsafe Image - Borderline Text - Borderline Combination
> * **U-S-B** stands for Unsafe Image - Safe Text - Borderline Combination
>
> So in both of these combinations the image is unsafe but the text is not clearly unsafe and therefore the joint label can be safe or borderline.
>
> ### Why Gemini-1.5?
>
> Gemini-1.5-Pro-002 was chosen for text generation throughout the paper as it was the most powerful model available to us at the time of data generation.
>
> ---
>
> We believe these substantial additions have significantly strengthened our work and addressed all concerns raised. The extensive new documentation provides full transparency while clarifying the nuanced nature of borderline cases and joint safety assessment. We hope the reviewer will find these improvements satisfactory and are happy to provide any additional clarifications needed.

---

> > ### Comment · Reviewer_L5ZW · 2025-11-26
> >
> > Thanks for the authors' clarifications. All my concerns have been adressed. I have raised my score accordingly

---

### Official Review · Reviewer_kBGh · 2025-11-01

**Soundness:** 2
**Presentation:** 2
**Contribution:** 2
**Rating:** 6
**Confidence:** 4

**Summary:**

The paper presents Vision Language Safety Understanding (VLSU), a comprehensive framework and benchmark to systematically evaluate safety in multimodal foundation models. This framework addresses the problem of joint interpretation, where individually safe image and text inputs become harmful in combination. The papers evaluation of current VLMs reveals a gap joint understanding. More detailed analysis indicate a Systematic over-sensitivity to any unsafe component and high error rate when combining modalities despite correct assessments on each individual one.

**Strengths:**

- the problem is well motivated and formulated. The fact that safety assignments should consider the interplay of both text and image inputs is intuitive and easy to follow
- introduces a comprehensive benchmark that will be a valuable contribution to the community
- safety taxonomy is grounded in prior work
- I appreciate the more nuanced three class classification over prevalent binary safe/unsafe
- Clear and significant findings on gaps in joint reasoning
- detailed error analysis in sections 4 and 5 on lack of model nuance, text modality dominance and error distribution

**Weaknesses:**

# Major

The major weakness of this paper is a significant lack of details on the construction methodology which also limits reproducibility
- further details on taxonomy guidelines (see below)
- **Stage 1 ** what is the exact setting for the "systematic parameterization"? what prompts where used and what were the inputs used in this parameterization?
- **Stage 2** There is no information provided on the image corpus used for retrieval. What is its size, origin, licensing? Is there some additional filters on image metadata wrt. resolution, aspect ratio, origins, etc? What exactly was the retrieval setting?
- ** Stage 3** Again no details on the synthetization pipeline are provided. What were the prompts and inputs?
- **Stage 4** no information on the makeup and setting of this human user study. Number of annotators, what makes them experts, compensation, instructions to annotators, measures to ensure that ethical guidelines where followed. What is the inter-annotator agreement?
- In addition to overall statistics of the dataset what are the distributions over categories and do we have label balance within each category as well?
-The paper should provide a datasheet for the newly introduced dataset (https://arxiv.org/abs/1803.09010) to adhere with standardized documentation practices, especially in safety related fields.


## Minor comments

**Presentation**
The presentation of the paper could be improved in parts. For one all Figure and Table captions are rather short making it hard to grasp for readers only skimming the paper. Especially, Figure 3 and 4 feel hard to grasp and would benefit from some additional information or change in visual presentation. Page 8 is very convoluted and the results in Sections 4 and 5--while interesting---are presented somewhat disjointed resulting in a lack of a common thread throughout the paper

**Human Oracle Topline**
Using a bootstrap of single annotators vs the majority consensus as human upper bound is more confusing to me then helpful. Since safety assessments are subjective to some extent deciding on a correct gold label per sample is challenging. Consequently, the human topline is much more an indicator of inter-annotator disagreement or the level of objectiveness then anything else. In my opinion the spread over inter-annotator agreement is a much more informative metric to plot and I would drop the human oracle.

**Taxonomy**
While the taxonomy described in A.2 seems reasonable it is lacking some important details. Compared to prior works like LlamaGuard or LlavaGuard a detailed description per category on what content makes for safe, borderline and unsafe content respectively would be helpful. This also ties in with lack of information on the human annotator setup. Where they provided with additional guidelines on the taxonomy or did they have to figure those out themselves?

**Reliance on GPT-4o**
The method uses GPT-4o to auto grade the severity of inputs. While hallucinations and inaccuracies here are mitigated by optimizing the system prompt agains a human annotated ground truth there might still be a selection bias here. However, since all final scores are human graded this is likely negligible, but the paper does not provide a qualitative assessment of GPTs accuracy on this pre-selection task.

**lack of qualitative examples**
In the Appendix the paper provideds 5 examples per safety level of the dataset but overall there is a significant lack of qualitative examples. for example, a comprehensive set of per-category examples and outputs of different models is missing.

**Limited Scope of Alignment Evaluation**
The safety alignment task (Section 4.4) only evaluates two models (Gemini-1.5 and Qwen2.5VL-32B) due to "compute constraints". While the findings are interesting (models trade off over-blocking for under-refusal based on prompts), this analysis feels incomplete. A broader evaluation across more models would be needed to claim this is a universal alignment gap.

**Questions:**

Please see the questions posed in the weakness section

**Details Of Ethics Concerns:**

The paper had human annotators label image-text pairs that systematically cover 15 harm categories across 17 safety pattern. However, no description is made of precautions taken to ensure annotator well-being. In general, the authors do not detail how human annotators where sourced, compensated, what the exact study setup looked like and misses details on the number and demographics of human annotators.
This is especially concerning given the sensitive nature of the content handled for this study.

---

> ### Author Response · Authors · 2025-11-21
> **Response to Reviewer Comments**
>
> We sincerely thank the reviewer for their detailed and thoughtful evaluation, which has substantially helped us improve the quality of our work.
> We have carefully addressed the concerns raised and have added extensive new content to the appendix, including complete methodological details, comprehensive annotation documentation, and a full datasheet.
>
> ## Responses to Major Weaknesses
>
> ### Stage 1: Systematic Parameterization Details
>
> We have now provided complete documentation of our image-concept generation pipeline:
>
> * **Appendix C.1**: We detail the exact parameterization including: (1) DSPy signature templates with full prompts verbatim, (2) the two-step generation process (basic concept generation followed by augmentation), (3) separate signatures for safe vs. unsafe image concepts, and (4) all input parameters (safety category, category description, number of queries).
>
> ### Stage 2: Image Corpus and Retrieval Details
>
> We have added comprehensive information about our image retrieval process:
>
> * **Appendix C.2**: We now specify: (1) corpus size: 3.6 billion images, (2) origin: publicly available images crawled by our web crawler respecting robots.txt directives, (3) licensing: will be released as URLs under CC-BY-NC-ND license, (4) no additional metadata filters were applied to maximize diversity, and (5) perceptual hashing-based de-duplication ensures each image appears only once in VLSU. Additional details that would disclose our affiliation can be provided during the camera-ready phase.
>
> ### Stage 3: Query Synthesis Pipeline Details
>
> We have provided exhaustive documentation of our context-driven query synthesis:
>
> * **Appendix C.3**: We detail: (1) the complete DSPy signature template, (2) all five text type instructions with exact prompts (binary questions, information-seeking, rhetorical, description, how-to) and (3) all three topic reference instructions mapping to severity levels (NO-MENTION→Safe, BENIGN-MENTION→Borderline, MALIGN-MENTION→Unsafe).
>
> ### Stage 4: Human Annotation Details
>
> We have significantly expanded our annotation documentation to address all concerns:
>
> * **Annotator Details (Appendix B.1)**: 225 expert annotators with 2+ years of experience; each sample annotated by 3 expert graders for text and combination labels; extensive training on 170-sample gold set with iterative guideline refinement.
>
> * **Ethical Considerations (Appendix B.2)**: Comprehensive ethical considerations including voluntary participation, ability to skip tasks, strict time limits, sensitive data training, image blur filters, 24/7 health resources, and direct communication channels with research team.
>
> * **Annotation Instructions (Appendix B.3)**: Complete annotation instructions with detailed per-category guidelines (Figures 11-15 show examples for safe, borderline, and unsafe content with specific examples for Drug Use harm category).
>
> * **Inter-Annotator Agreement (Appendix B.4)**: Inter-annotator agreement was computed as overall Krippendorff's alpha: 0.806 and overall raw agreement: 0.869. Additional raw agreement numbers are broken down by all 17 severity combinations in Table 9.
>
> ### Dataset Distributions and Balance
> We have added comprehensive dataset statistics showing that our dataset is very well balanced across harm categories and severity combinations (*Appendix A.3**: Figure 9, 10 and Table 6, 7)
>
> ### Datasheet
>
> We have included a complete standardized datasheet following Gebru et al. (2021) in Appendix F.
>
> ## Responses to Minor Comments
>
> ### Human Oracle Topline
> We agree that raw inter-annotator agreement offers additional valuable information and have added detailed breakdown in Appendix B.4, Table 9. The human oracle provides an upper bound for model performance but we will consider removing or reframing in the camera-ready version.
>
> ### Taxonomy Guidelines
> We have added extensive per-category description in Appendix A1, Table 7.
>
> ### GPT-4o Pre-selection Bias
> We optimized the GPT-4o judge using DSPy on a 170-sample gold set with 7 expert annotations, and only used it for initial filtering to 10,000 samples before human annotation. Since all final labels are human-graded with high inter-annotator agreement (α=0.806), we believe any potential pre-selection bias is mitigated.
>
> ### Qualitative Examples
>
> We have added qualitative examples for all severity levels in Appendix A.2, Figures 6-8.
>
> ---
>
> Overall these substantial additions have significantly strengthened our work and addressed all major concerns raised. The extensive new documentation in the appendix provides full transparency and we hope the reviewer will reconsider their assessment in light of this new evidence. We are happy to provide any additional clarifications needed.

---

> ### Author Response · Authors · 2025-11-21
> **Addressing Ethics Concerns**
>
> We take the ethics concerns very seriously and have provided comprehensive documentation:
>
> ## Annotator Sourcing and Compensation:
>
> * **Appendix B.1**: All 225 annotators are full-time corporate employees in the United States and European Union receiving competitive compensation and full benefits for this work. Annotator quality is maintained via extensive certifications and documented performance scores. All annotators are proficient English speakers.
>
> ## Annotator Well-being Protections:
>
> * **Appendix B.2**: We implemented comprehensive safeguards including: (1) completely voluntary participation with ability to opt out at any time with no effect on employment or performance rating, (2) ability to skip any individual task, (3) strict time limits on exposure to sensitive data per day, (4) comprehensive sensitive data training covering potential risks, (5) internal tools to minimize exposure (e.g., image blur filters), (6) 24/7 access to health and well-being resources, and (7) direct communication channels with the research team.
>
> ## Study Setup and Demographics:
>
> * **Appendix B.1**: We do not collect additional demographic information beyond language proficiency because extensive quality control (certifications, performance scores, training) should minimize annotator bias. This decision balances privacy concerns with quality assurance.
>
> ## Annotation Instructions:
>
> * **Appendix B.3**: Complete annotation guidelines with detailed per-category instructions, examples, and explicit definitions of safe/borderline/unsafe content to ensure consistent and informed judgments.
>
> ---
>
> We believe these comprehensive protections demonstrate our commitment to responsible research practices and annotator welfare.

---

### Official Review · Reviewer_DnZe · 2025-11-01

**Soundness:** 1
**Presentation:** 2
**Contribution:** 1
**Rating:** 2
**Confidence:** 4

**Summary:**

This paper presents VLSU, a systematic vision–language safety framework and a benchmark of 8,187 real image–text pairs that captures cases where individually benign modalities become harmful when combined, and empirically demonstrates that models fail dramatically at compositional multimodal understanding. The authors introduce a Borderline class, the design of 17 combinatorial severity patterns, and a multi-stage pipeline that combines automatic filtering with human annotation.

**Strengths:**

1.	The inclusion of a borderline safety category is meaningful, and its necessity points to an appropriate direction for this field.

2.	The curation of 8,187 human-annotated real-image pairs, each categorized across harm types and joint safety patterns, is a significant step towards realistic, actionable safety evaluation.

**Weaknesses:**

1.	This paper does not contain several important related works that address multimodal safety evaluation benchmark, a key topic of this paper. A comparison with the following papers is necessary: ELITE [1], VLGuard [2], MLLMGuard [3], JailbreakV-28k [4]

2.	There is insufficient information about the human annotators. The paper states, "The image grade is labeled by one senior expert grader", which could introduce bias. Furthermore, the criteria for human annotators to judge "borderline" cases are ambiguous. Reliability could be improved by providing more specific guidelines and ensuring diversity among the annotators. This point is critically related to the overall reliability of the dataset.

3.	In lines 65-67, the issue of an image and text being individually safe but suggesting self-harm intent when combined has already been discussed in SIUO [5]. Although SIUO is cited in the related work section, it should also be cited in the Introduction.

4.	Adding qualitative examples for borderline cases would help understand the paper.

[1] Wonjun Lee, Doehyeon Lee, Eugene Choi, Sangyoon Yu, Ashkan Yousefpour, Haon Park, Bumsub Ham, and Suhyun Kim. ELITE: Enhanced language-image toxicity evaluation for safety. In Forty- second International Conference on Machine Learning, 2025.

[2] Zong, Y., Bohdal, O., Yu, T., Yang, Y., and Hospedales, T. Safety fine-tuning at (almost) no cost: A baseline for vision large language models. In Forty-first International Conference on Machine Learning , 2024.

[3] Gu, T., Zhou, Z., Huang, K., Dandan, L., Wang, Y., Zhao, H., Yao, Y., xingge qiao, wang, K., Yang, Y., Teng, Y., Qiao, Y., and Wang, Y. MLLMGuard: A multi-dimensional safety evaluation suite for multimodal large language models. In The Thirty-eight Conference on Neural Infor- mation Processing Systems Datasets and Benchmarks Track, 2024.

[4] Luo, W., Ma, S., Liu, X., Guo, X., & Xiao, C. Jailbreakv: A benchmark for assessing the robustness of multimodal large language models against jailbreak attacks. First Conference on Language Modeling, 2024

[5] Wang, S., Ye, X., Cheng, Q., Duan, J., Li, S., Fu, J., ... & Huang, X. Safe Inputs but Unsafe Output: Benchmarking Cross-modality Safety Alignment of Large Vision-Language Model. Findings of the Association for Computational Linguistics: NAACL 2025

**Questions:**

1.	Could the authors provide a comparison with the other benchmarks mentioned (e.g., ELITE, VLGuard)? This would significantly help in establishing the reliability and contribution of VLSU.

2.	Could you provide the per-category inter-annotator agreement and Pearson correlation among the annotators?

---

> ### Author Response · Authors · 2025-11-21
>
> We sincerely thank the reviewer for their thorough evaluation and constructive feedback, which has substantially helped us improve the quality of our work. Below, we have carefully addressed all the concerns raised and have added significant new content to the appendix.
>
> ## Questions
>
> ### Q1: Could the authors provide a comparison with the other benchmarks mentioned (e.g., ELITE, VLGuard)?
>
> Thank you for this important suggestion and highlighting important existing work. We have added the ELITE, VLGuard and MLLMGuard benchmarks to our comprehensive comparisons:
>
> * **Extended benchmark comparison (Appendix E.1, Table 11)**: We have added results for ELITE, VLGuard, and MLLMGuard alongside VLSU and existing benchmarks. Our results show that VLSU remains substantially more challenging than existing benchmarks with best models achieving 95.9% F1 on ELITE, 87.8% on VLGuard and 86.9% on MLLMGuard compared to the 70.9% F1 on VLSU, demonstrating VLSU's unique ability to expose joint understanding limitations. We keep this information in the Appendix for the rebuttal period to clearly delineate the added material but will move it into the main paper for camera ready.
>
> * **Detailed dataset comparison (Appendix A.4, Table 8)**: We additionally provide a comprehensive comparison between VLSU and existing datasets, showing that VLSU is the first dataset to provide: (1) comprehensive coverage with borderline data across 17 cross-modal severity combinations, (2) large-scale real images not pooled from existing datasets, and (3) rigorous human annotations on image, text, and their combination modalities.
>
> We did not additionally include JailbreakV-28k since the focus of our dataset is not on targeted, adversarial inputs but rather natural inputs with nuanced variations in their safety properties.
>
> ### Q2: Could you provide the per-category inter-annotator agreement and Pearson correlation among the annotators?
>
> We have added detailed inter-annotator agreement analysis to Appendix B.4:
>
> * For the whole dataset, we find a Krippendorff's alpha coefficient of 0.806, indicating good inter-annotator agreement. Given our annotation setup with categorical labels and 3 annotators being sampled randomly for each datapoint from a larger pool of 225 graders we find this to be a more appropriate statistic to compute instead of the requested Pearson correlation.
>
> * **Appendix B.4, Table 9**: We now report pairwise raw agreement scores broken down by all 17 VLSU severity combinations with 95% confidence intervals. Agreement ranges from 0.541 (B-B-U) to 0.945 (B-S-B), indicating the more challenging nature of certain severity combinations. Nevertheless even a ~54% annotator agreement is substantially larger than chance agreement which is 1/9 = ~11% only considering the Safe, Borderline, Unsafe label options and 1/16 = ~6% when including the additional 'not sure' option available to annotators.
>
> ## Additional Concerns
>
> ### Weakness 1: Missing related work comparisons
>
> As detailed in our response to Q1, we have now included comprehensive comparisons with ELITE, VLGuard and MLLMGuard [3] in our extended results (Appendix E.1).
>
> ### Weakness 2: Insufficient information about human annotators
>
> We appreciate this concern and have significantly expanded our annotation documentation:
>
> * **Appendix B.1-B.3**: We now provide comprehensive details about our annotation process, including: (1) 225 expert annotators with 2+ years of experience, (2) each sample annotated by 3 expert graders for text and combination labels, (3) extensive training on a 170-sample gold set, and (4) detailed annotation guidelines with per-category instructions (Figures 11-15).
>
> * **Image annotation**: While image labels were assigned by one author with extensive safety annotation experience, we note that: (1) the critical joint understanding evaluation relies on the combination labels which have 3-annotator agreement, and (2) our error analysis (Section 5, Figure 5) shows that 34% of combination errors occur despite correct image classification, demonstrating that image labeling is not the bottleneck.
>
> * **Borderline criteria**: We provide explicit definitions and extensive examples in our annotation guidelines (Appendix B.3, Figures 12, 14-15), showing annotators how to distinguish borderline content that "references unsafe topics but in educational or informational context" from clearly unsafe content.
>
> ### Weakness 3: Citation of SIUO in Introduction
>
> We added the SIUO citation in the Introduction alongside our discussion of jointly harmful content in the updated version of the manuscript.
>
> ### Weakness 4: Qualitative examples for borderline cases
>
> We have added extensive qualitative borderline examples in **Appendix A.2, Figure 6**.
>
> ---
>
> We believe these substantial additions have significantly strengthened our work and hope the reviewer will reconsider their assessment in light of this additional material. We are happy to provide any additional clarifications needed.

---

> > ### Comment · Reviewer_DnZe · 2025-11-26
> >
> > I would like to thank the authors for their rebuttal.
> >
> > However, I still have a few remaining concerns that I would like the authors to address. I appreciate the inclusion of the inter-annotator agreement analysis in Appendix B.4. However, the reported agreement scores raise a new concern regarding the reliability of the "Borderline" labels. According to the rebuttal, the pairwise raw agreement ranges from 0.541 to 0.945. Specifically, several categories exhibit agreement rates below 0.65, suggesting that annotators frequently disagreed on these labels.
> >
> > Does the low agreement in Borderline-related categories stem from ambiguous annotation guidelines, or is it due to the inherent subjectivity of "Borderline" content? If it is the latter, how can we justify using a single "ground truth" label for evaluation when even human experts lack consensus?

---

> > > ### Author Response · Authors · 2025-12-03
> > >
> > > Thank you for this important follow-up question about annotation reliability, particularly for borderline categories. We appreciate the opportunity to clarify this critical aspect of our dataset.
> > >
> > > **Borderline labels show strong agreement overall:**
> > >
> > > While we acknowledge that safety/harmfulness assessment is inherently subjective, we don't observe systematic degradation in agreement for borderline content. Across all combined labels, our agreement rates are: Safe: 0.965 (6,590 pairs), Borderline: 0.877 (9,622 pairs), Unsafe: 0.776 (7,879 pairs). So while there are some subcategories with lower pairwise agreement, overall the borderline class as well as the safe and unsafe categories show fairly high agreement suggesting annotators can reliably identify these categories.
> > >
> > > **Agreement rates indicate reliable annotation:**
> > >
> > > Even ~60% pairwise agreement demonstrates significant consensus when properly contextualized:
> > >
> > > (A) With 3 annotators per sample, if 2 agree and 1 diverges, pairwise agreement is only 33%. Thus, for our lowest-agreement category (B-B-U at 54.1%), we have 24 samples with complete 3-way agreement and 53 samples with 2/3 majority agreement, indicating substantial consensus.
> > >
> > > (B) Consider 5 annotators (10 pairs): if 4 fully agree and 1 disagrees, this yields only 60% pairwise agreement (6 agreeing pairs, 4 disagreeing). Hence, 60% agreement actually reflects strong annotator consensus.
> > >
> > > **Dataset remains valuable even with conservative filtering:**
> > >
> > > Importantly, 6,789 samples (81.4% of the dataset) achieved full annotator agreement while still covering all 17 severity combinations. Additionally, we will release individual annotations, enabling researchers to filter samples by agreement level or weigh them accordingly for their specific use cases.
> > >
> > > While we acknowledge the inherent subjectivity in safety assessment, our agreement metrics are consistent with reliable annotation processes, and we believe the dataset provides substantial value to the community through both its scale and granular annotation data.

---

### Author Response · Authors · 2025-12-03
**Summary for Area Chair: VLSU Rebuttal Discussion**

We thank all the reviewers for their feedback on our work that has significantly strengthened it and to the area chairs for their efforts. Below we summarize each reviewer's primary concerns and our responses to provide a concise overview of the rebuttal discussion.


### **Score Improvements**

1. Reviewer DnZe:  Acknowledged comprehensive additions addressed all original concerns; engaged in constructive follow-up discussion about borderline annotation reliability
1. Reviewer kBGh: No response to rebuttal
1. Reviewer L5ZW: **score increase from 6 → 8**. Explicitly confirmed "all concerns addressed" and raised score accordingly
1. Reviewer RKh7: **score increase from 4 → 6**. Satisfied with responses and improvements



### **Key Additions During Rebuttal**

We substantially expanded the paper with new content addressing all reviewer concerns:

1. **Additional benchmark comparisons** with ELITE, VLGuard, MLLMGuard (Appendix E.1, Table 11)
1. **Additional model evaluations** on GLM4.1V (Appendix E.1, Table 11)
1. **Complete annotation documentation** including expert annotators details, guidelines, and inter-annotator agreement (Krippendorff’s α=0.806) in Appendix B.
1. **Full methodological details** for all dataset construction stages in Appendix C.
1. **Extensive qualitative examples** of borderline cases (Figures 6, 7, 8 in Appendix A.2)
1. **Complete datasheet** following Gebru et al. (2021) in Appendix F.



### **Reviewer-by-Reviewer Summary**

1. Reviewer DnZe
    - **Main concern:** Comparison with three additional benchmarks (ELITE, VLGuard and MLLMGuard) and inter-annotator agreement details.
    - **Response:** We added evaluations for all three datasets and find VLSU remains substantially more challenging than all other datasets. Inter-annotator agreement using Krippendorff’s alpha is 0.806 for the full dataset, with high agreement for all three safe, borderline and unsafe categories, indicating good inter-annotator agreement.
2. Reviewer kBGh
    - **Main concern:** Lack of details on the dataset construction methodology and request to include datasheet.
    - **Response:** We added a detailed dataset construction methodology to Appendix C covering all settings and prompts for data generation, details of image dataset used to retrieve images and complete human annotation guidelines. We also included a detailed Datasheet as requested by the reviewer (Appendix F).
3. Reviewer L5ZW
    - **Main concern:**  Requested more details about dataset construction, language diversity and annotation guidelines especially for Borderline class.
    - **Response:** We provided explicit operational definitions for the borderline class (Appendix B.3) and detailed dataset construction methodology in Appendix C. We acknowledged VLSU is English-only and added this as a limitation with future multilingual work as an exciting direction.
4. Reviewer RKh7
    - **Main concern:** Clarification of how VLSU goes beyond existing datasets
    - **Response:** We added a detailed comparison between VLSU and existing multimodal safety datasets in Appendix A.4, Table 8, highlighting fundamental contributions like borderline severity class, systematic compositional framework and scale and realism. We also evaluated on another reasoning model GLM-4.1V upon reviewers request and find the trends hold across models.



### **Ethics Concerns Resolution**

All ethics concerns were comprehensively addressed:
- Annotator welfare: Voluntary participation, ability to skip tasks, time limits, 24/7 health resources
- Copyright/privacy: Public images via URLs, respects robots.txt, CC-BY-NC-ND license
- Documentation: Complete annotator sourcing, compensation, and safety protocols

---

### Meta-Review · Area_Chair_nT8f · 2026-01-10

**Summary:**

The paper introduces VLSU, a new benchmark for joint vision-language safety. The dataset consists of 8,187 real image-text pairs with15 harm categories, 17 combinatorial severity patterns and human annotations.

Strengths

- The dataset focuses on joint multimodal safety rather than unimodal safety, which is currently underexplored in the literature.

- It includes human-annotated, realistic image–text pairs.

- The inclusion of borderline cases enables evaluation of both over-refusal and under-refusal behaviors.

- The paper evaluates a wide range of open- and closed-source vision–language models.

Weaknesses raised by the reviewers

- Labels for borderline cases may be subjective.

- Missing comparisons with existing benchmarks.

- Insufficient details on dataset construction and the human annotation process.

During the rebuttal, the authors provide comprehensive responses, including comparisons with other benchmarks and clarifications of dataset construction and annotation details, which address most of the concerns. Overall, this work represents a timely contribution to multimodal AI safety.

**Reviewer Concerns:**

Addressed:

- Labels for borderline cases may be subjective.

- Missing comparisons with existing benchmarks.

- Insufficient details on dataset construction and the human annotation process.

Unaddressed:

The previous AC has flagged this submission for ethical review. Given that the paper introduces a benchmark containing harmful image-text pairs and relies on human annotation, I agree that an ethical review would be appropriate. However, I did not find an ethical review, so I'm not sure if this is addressed.

**Reviewer Scores:**

They will likely raise scores

---

### Decision · Program_Chairs · 2026-01-26

Accept (Poster)